# Redox-Active Cerium Fluoride Nanoparticles Selectively Modulate Cellular Response against X-ray Irradiation In Vitro

**DOI:** 10.3390/biomedicines12010011

**Published:** 2023-12-20

**Authors:** Nikita N. Chukavin, Kristina O. Filippova, Artem M. Ermakov, Ekaterina E. Karmanova, Nelli R. Popova, Viktoriia A. Anikina, Olga S. Ivanova, Vladimir K. Ivanov, Anton L. Popov

**Affiliations:** 1Institute of Theoretical and Experimental Biophysics, Russian Academy of Sciences, Pushchino 142290, Russia; chukavinnik@gmail.com (N.N.C.); kristina.kamensk@mail.ru (K.O.F.); ao_ermakovy@rambler.ru (A.M.E.); silisti@bk.ru (E.E.K.); nellipopovaran@gmail.com (N.R.P.); viktoriya.anikina@list.ru (V.A.A.); 2Scientific and Educational Center, State University of Education, Moscow 105005, Russia; 3Frumkin Institute of Physical Chemistry and Electrochemistry, Russian Academy of Sciences, Moscow 119071, Russia; runetta05@mail.ru; 4Kurnakov Institute of General and Inorganic Chemistry, Russian Academy of Sciences, Moscow 119991, Russia; van@igic.ras.ru

**Keywords:** cerium fluoride nanoparticles, DNA reparation, double-strand brakes, X-ray radiation

## Abstract

Ionizing radiation-induced damage in cancer and normal cells leads to apoptosis and cell death, through the intracellular oxidative stress, DNA damage and disorders of their metabolism. Irradiation doses that do not lead to the death of tumor cells can result in the emergence of radioresistant clones of these cells due to the rearrangement of metabolism and the emergence of new mutations, including those in the genes responsible for DNA repair. The search for the substances capable of modulating the functioning of the tumor cell repair system is an urgent task. Here we analyzed the effect of cerium(III) fluoride nanoparticles (CeF_3_ NPs) on normal (human mesenchymal stem cells–hMSC) and cancer (MCF-7 line) human cells after X-ray radiation. CeF_3_ NPs effectively prevent the formation of hydrogen peroxide and hydroxyl radicals in an irradiated aqueous solution, showing pronounced antioxidant properties. CeF_3_ NPs are able to protect hMSC from radiation-induced proliferation arrest, increasing their viability and mitochondrial membrane potential, and, conversely, inducing the cell death of MCF-7 cancer cells, causing radiation-induced mitochondrial hyperpolarization. CeF_3_ NPs provided a significant decrease in the number of double-strand breaks (DSBs) in hMSC, while in MCF-7 cells the number of γ-H2AX foci dramatically increased in the presence of CeF_3_ 4 h after irradiation. In the presence of CeF_3_ NPs, there was a tendency to modulate the expression of most analyzed genes associated with the development of intracellular oxidative stress, cell redox status and the DNA-repair system after X-ray irradiation. Cerium-containing nanoparticles are capable of providing selective protection of hMSC from radiation-induced injuries and are considered as a platform for the development of promising clinical radioprotectors.

## 1. Introduction

Ionizing radiation, which is commonly used in cancer therapy, is electromagnetic radiation or photons that ionize the molecules in the cell [1]. Ionized molecules are highly reactive species undergoing a rapid cascade of chemical transformations leading to the breaking of chemical bonds [2]. As a result, the DNA structure, the main target of ionizing radiation, is disrupted, and highly reactive products are formed from water molecules contained in the cell, such as reactive oxygen species (ROS) and free radicals [3]. Consequently, exposure to ionizing radiation directly and indirectly leads to a damage to the basic structures of the cell and its death. However, ionizing radiation also triggers a series of signaling cascades that promote cell survival and activate cellular processes such as DNA repair, cell cycle arrest and autophagy, which leads to radioresistance and the development of tumor cell adaptability [4].

The selective increase in radiosensitivity of tumor cells is one of the important tasks of the development of a new modern treatment methods of solid tumors radiation therapy [5]. There are a few approaches to increase the radiation sensitivity of cancer cells, starting with increasing the level of tumor tissue oxygenation, ending with the use of kinase inhibitors (for example, PI3K-like kinases) or cell cycle checkpoints [6]. The most promising agents for these purposes is the use of nanoradiosensitizers based on high Z elements, which can effectively accumulate in tumor cells and locally enhance the radiation dose, thereby increasing the damaging effect of ionizing radiation [7,8]. Some of them have already been approved by the FDA or are at various stages of clinical trials [9,10]. Thus, the use of nanoscale particles with a good biosafety profile and the ability to effectively absorb ionizing radiation is a promising strategy in the development of new effective radiosensitizers.

Fluoride nanoparticles have their unique physicochemical properties, which determines such interest in the study. In particular, lanthanide fluorides (LnF_3_, Ln = lanthanides) have very-low solubility (about 10^−5^–10^−6^ mol/L) [11] and, as a consequence, low toxicity [12]. For example, previous studies in vivo on planarians confirm that CeF_3_ nanoparticles have low toxicity at concentrations of 1 μM–10 mM [13]. Cerium fluoride is one of the best scintillators for use in biomedicine [14]. Scintillators are compounds that are capable of emitting photons when they absorb ionizing radiation of various types, including X-ray. CeF_3_ NPs is an effective scintillator, since when irradiated with X-ray, it emits UV light due to fluorescence at a wavelength of 325 nm. In the nanoscale state, cerium ions are able to transfer excitation energy to ions in its doped crystal lattice, for example, rare-earth ions of other metals (for example, europium or terbium) located in the crystal lattice [15]. In [16], the cytotoxicity of Ce^3+^, Tb^3+^: NaYF_4_ spherical particles against Capan-1 cells with a size of 16.7 nm was studied in the concentration range of 0.5 pM–40 nM using the MTT assay. After 48 h of incubation, a dose-dependent linear decrease in viability was detected from ~100% at 0.5 pM to ~30% at 40 nM. According to the general opinion, cerium-containing nanoparticles with a predominant content of cerium in the trivalent state on the surface provide their SOD-mimetic activity [17]. Ce^3+^ ions cause an apoptosis decrease through their redox activity. In particular, Ce^3+^-rich cerium dioxide nanoparticles stabilized by PEG secure cells from cell death induced by ROS in vitro as well as decrease the frequency of ischemic cell death in vivo [18]. Soh and co-authors synthesized Zr-doped cerium dioxide nanoparticles with a size of 2 nm and a high content of Ce^3+^ ions, which reduced inflammation in different representative sepsis models [19]. Additionally, it has been shown that both unmodified and Tb-doped CeF_3_ nanoparticles exhibit effective antivirus activity in vitro [12]. CeF_3_ NPs were shown to prevent oxidative discoloration of indigo carmine dye induced by exposure to hydrogen peroxide, where a dose-dependent protective effect was detected. Luminescent nanocrystals containing lanthanide Ln^3+^ ions are currently of interest to researchers due to their specific properties such as high physico-chemical sustainability, resistance to both photobleaching and photochemical decomposition as well as a long lifetime of luminescence [20,21,22]. Additionally, such lanthanide nanocrystals have the property of forming colloidal solutions with high stability [23]. CeF_3_:Tb@LaF_3_-based nanomaterials are promising biomedical agents due to their high colloidal stability and resistance to photobleaching [24,25]. CeF_3_:Tb^3+^ nanoparticles coated with lanthanum (III) fluoride shells increase the intensity of emission by more than 25% in comparison with unmodified terbium-doped cerium(III) fluoride nanoparticles. Lanthanum (III) fluoride shells provided a barrier for transfer of energy resulting in significant decrease in energy loss from the surface centers of luminescence. Consequently, the luminescence lifetime and intensity of modified nanoparticles have significantly increased [26]. The luminescent and magnetic properties of lanthanide nanoparticles may be of fundamental importance in the development of multifunctional nanomaterials, allowing for them to be used in MRI and other luminescence methods [27,28]. CeF_3_ NPs can be considered as an effective platform for development of promising redox-active drugs that can find their application in the therapy and diagnosis of oncological diseases.

In this work, redox-active CeF_3_ NPs were used to analyze their impact to cell proliferation, mitochondrial membrane potential and DNA-repair process upon X-ray irradiation. The effect CeF_3_ NPs on the rate of DSBs in normal and cancer cells was analyzed and the possible molecular mechanisms of this effect were discussed. The significant improvement in the DSBs DNA-repair rate by CeF_3_ NPs in normal cells exposed to ionizing radiation shows new opportunities in development of advanced radiation-mediated cancer therapies.

## 2. Materials and Methods

### 2.1. Synthesis and Characterization of CeF_3_ NPs

CeF_3_ NPs were synthesized by precipitation in alcoholic media [12]. UV–vis absorption spectra of the colloid solution was recorded using standard quartz cells for liquid samples on a UV5 Nano spectrophotometer (Mettler Toledo, Columbus, OH, USA). Transmission electron microscopy and selected area electron diffraction (SAED) analysis was performed using a Leo 912 AB Omega electron microscope operating at 100 kV acceleration voltage. Nanoparticle size measurements by dynamic light scattering and zeta-potential were carried out on a BeNano Zeta particle size analyzer (BetterSize, Dandong, China). Powder XRD (X-ray powder diffraction analysis) of cerium(III) fluoride nanoparticles was performed using a Bruker D8 Advance diffractometer (CuKα radiation) (Billerica, MA, USA) with a signal acquisition time of 0.4 s/step and a step of 0.02°2Θ in the angle range of 20–90°2Θ. XRD patterns full-profile analysis was performed using TOPAS software v.4.2 (Billerica, MA, USA) and diffraction maxima were fitted to pseudo-Voigt functions.

### 2.2. Cell Cultures

The experiments were performed using a culture of human mesenchymal stem cells (hMSC) derived from the third molar bud, extracted from a healthy 12-year-old patient for orthodontic indications, and MCF-7 human breast cancer cell line, obtained from Theranostics and Nuclear Medicine Laboratory cryostorage (ITEB RAS, Pushchino, Russia). The third molar bud was removed according to the orthodontic indications of the dental clinic «Dr. MUN» (Moscow, Russia) in accordance with the ethics committee after consent was signed by the patient’s parents. The in vitro experiments were performed with the permission of the Ethics Committee of Institute of Theoretical and Experimental Biophysics of the Russian Academy of Sciences (ITEB RAS, Pushchino, Moscow, Russia) in accordance with the protocol No. 35 dated 5 March 2022 and in accordance with good clinical practice and the ethical principles of the current edition of the Helsinki Declaration. The cells were cultured in a DMEM/F12 (1:1) culture medium containing 50 μg/mL of penicillin, 50 μg/mL of streptomycin, 10% of fetal bovine serum (FBS) and 1% of l-glutamine at a temperature of 37 °C in a 95% humidity atmosphere containing 95% air and 5% CO_2_. The cells were seeded on 96-well and 6-well plates at a density of 20,000–35,000/cm^2^. The culture medium was then replaced with a fresh culture medium containing cerium(III) fluoride nanoparticles in different concentrations (10^−3^–10^−7^ M) 6 h after cell attachment. Cells from the control groups were cultured without the addition of CeF_3_ NPs.

### 2.3. X-ray Irradiation

X-ray irradiation of the cell cultures was performed using an X-ray therapeutic machine RTM-15 (Mosrentgen, Moscow, Russia) in a dose of 1 and 5 Gy for CeF_3_ NPs solution, 15 Gy for cell cultures, 1.5 Gy for DSBs analysis at a dose rate of 1 Gy/min, 200 kV voltage, 37.5 cm focal length and a 20 mA current.

### 2.4. Assessment of CeF_3_ NPs Antioxidant Activity

To evaluate the antioxidant activity of CeF_3_ NPs, we determined the concentration of hydrogen peroxide after X-ray irradiation of CeF_3_ NPs by enhanced chemiluminescence using a luminol–4-iodophenol–peroxidase system [29]. We used TRIS buffer to maintain constant pH 7.2. The irradiation dose was 5 Gy (1 Gy per min). A liquid scintillation counter Beta-1 (MedApparatura, Kiev, Ukraine) was used as a highly sensitive chemiluminometer. This liquid scintillation counter was operating in the counting single photons mode with one photomultiplier and the coincidence scheme disengaged. The use of a Beta-1 liquid scintillation counter allows for registering hydrogen peroxide at a concentration of <1 nM. The hydrogen peroxide content was determined using calibration dependencies of chemiluminescence on the hydrogen peroxide concentration in solution. The hydrogen peroxide concentration used for calibration was measured spectrophotometrically at an absorbance wavelength of 240 nm using ε = 43.6 M^−1^ × cm^−1^. The hydroxyl radical formation under X-ray irradiation was measured using coumarin-3-carboxylic acid [30]. The fluorescence intensity of hydroxylated coumarin-3-carboxylic acid (ex 395/em 450 nm) was measured on a Cary Eclipse spectrofluorimeter (Agilent, Santa Clara, CA, USA) at room temperature.

### 2.5. Cell Proliferation Analysis

The cell growth analysis after incubation with CeF_3_ NPs and X-ray irradiation was estimated by counting the number of the cells stained with Hoechst 33342 using fluorescent imager Zoe^®^ (BioRAd, Hercules, CA, USA). At least four measurements were made for each CeF_3_ NPs (10^−3^–10^−7^ M). The data are presented as growth curves (mean with SD).

### 2.6. Analysis of DNA Double-Strand Breaks by γ-H2AX Foci

Cells were seeded in 35 mm Petri dishes with a central hole (Ibidi, Gräfelfing, Germany). After X-ray irradiation, cells were washed twice using PBS buffer solution (phosphate-buffered saline, pH 7.4, 135 mM NaCl, 0.5 mM KH_2_PO_4_, 3.2 mM Na_2_HPO_4_, 1.3 mM KCl). Subsequently, cells were fixed using PBS with paraformaldehyde (4%) for 15 min, permeabilized using PBS with Triton X-100 (0.1%) for 5 min, washed extensively with PBS buffer solution and blocked in PBS with BSA (5%) (Gibco, Paisley, Scotland) for 30 min at room temperature. Recombinant monoclonal gamma H2A.X (phospho S139) antibody conjugated to Alexa Fluor^®^ 488 (Abcam, Waltham, Boston, MA, USA) were used for immunostaining. Image acquisition of the DSBs was performed using an inverted fluorescent microscopy Zeiss Axiovert Observer 200 (Carl Zeiss Microscopy, Jena, Germany). Analysis of γH2AX stained cells was performed with an optional manual correction in individual cells. For each experimental group, at least 200 cells were analyzed. Image J software (Version: 1.53t) was used to quantify the detected γH2AX foci.

### 2.7. qPCR (Gene Expression Analysis)

RNA isolation from cell cultures was performed using innuPREP RNA Mini Kit 2.0 (Innuscreen, Berlin, Germany) according to the manual of the manufacturer. The gene-specific primers (Appendix A) were designed using Primer-BLAST tool (NCBI). First strand cDNA samples were synthesized using isolated from cell cultures RNA samples using an MMLV RT kit (Evrogen, Moscow, Russia) according to the manual of the manufacturer. qPCR was performed using synthesized first strand cDNA samples, designed primers for selected genes and qPCR-mix with SYBR Green I and ROX (Evrogen, Moscow, Russia) on a BioRad CFX-96 amplifier. Gene expression levels were normalized by the housekeeping gene expression levels (including rplp0, β-actin, and GAPDH). Gene expression changes were analyzed relative to the values of control groups. Genesis software (Version 1.8.1) was used to build gene expression heat maps and to perform principal component analysis (PCA) of the results.

### 2.8. Statistical Analysis

The experiments were conducted in 3–4 repetitions, with three independent repetitions for each CeF_3_ NPs concentration. Experimental results were compared with untreated control. Statistical analysis was performed using the methods of variation statistics (ANOVA, Student’s *t*-test). The mean values and the standard deviation (SD) of the mean were determined. The obtained data were processed using the GraphPad 8.0 Software.

## 3. Results and Discussion

The synthetic procedures resulted in stable aqueous sol of highly crystalline CeF_3_ NPs. Selected area electron diffraction patterns of the samples (Figure 1a) which correspond to the *P6_3/mcm* space groups for hexagonal CeF_3_ structure. High crystallinity of CeF_3_ NPs synthesized using soft chemistry routes is typical for this Ce(III)-containing material [12]. According to XRD results, the resulting sol contains as a dispersed phase pure hexagonal cerium fluoride (sp. gr. P63/mcm) with a crystallite size of 24 ± 2 nm (Figure 1b). ζ-potential of CeF_3_ NPs sol equals +41 mV, corroborating its high stability. The hydrodynamic radii of CeF_3_ NPs in the sol diluted with distilled water was 40 ± 2 nm (Figure 1c), indicating the low degree of the particle agglomeration. According to TEM data, the size of CeF_3_ NPs was 15–25 nm. The nanoparticles possess notable faceting, which indicates a high degree of crystallinity. Highly crystalline CeF_3_ NPs is expected to contain only trivalent cerium [31,32]. The main UV absorption peak of CeF NPs is located at 250 nm (Figure 1d). According to the literature data, UV absorption spectra of Ce^3+^ ions have a maximum at 253.6 nm with a molar extinction coefficient of 685 M^−1^ cm^−1^ [33]. The UV spectra of CeF_3_ NPs after treatment with H_2_O_2_ confirms the change in the valence of cerium ions after the reaction (Appendix A). Data presented in Figure 1e shows that CeF_3_ NPs are able to neutralize hydrogen peroxide formed in water upon X-ray exposure due to their catalase-mimetic properties. In the presence of CeF_3_ NPs at a concentration of 10^−6^ M, a 30% decrease in the hydrogen peroxide concentration was observed. Moreover, CeF_3_ NPs at a concentration of 10^−5^ M decreased the hydrogen peroxide level to zero values after X-ray irradiation. Such CeF_3_ NPs’ catalase-like activity can be explained by the number of hydrogen peroxide binding sites, which directly related to the size of these nanoparticles. Hydrogen peroxide catalytic decomposition is a heterogeneous process that occurs at the nanoparticles interface and depends on their specific surface area. CeF_3_ NPs have a size of 15–25 nm, which corresponds to ≈10% of the surface cerium atoms [34,35]. Thereby, the CeF_3_ NPs’ specific surface area occupies a smaller part of its entire surface. CeF_3_ NPs reduce the concentration of hydroxyl radical (Figure 1f), which is formed during irradiation only in high concentrations (10^−5^ M).

CeF_3_ NPs selectively modulate the proliferative activity of normal and tumor cells (Figure 2). Preincubation of hMSC with CeF_3_ NPs leads to an increase in their proliferative activity (Figure 2a,b). Irradiation of hMSC leads to inhibition of proliferative activity (reduction to 40% relative to non-irradiated control); however, cell pretreatment with CeF_3_ NPs ensures the preservation of positive dynamics of proliferative activity at concentrations (10^−3^–10^−4^ M), which confirms the pronounced radioprotective effect. At the same time, the proliferative activity of tumor cells significantly decreases in the presence of CeF_3_ NPs on the 5th day of coincubation, and X-ray irradiation (15 Gy dose) enhances this effect (Figure 2c,d). Thus, CeF_3_ NPs, on the one hand, stimulate the proliferative activity of normal cells, and, on the other, suppress the proliferative activity of tumor cells and have a radiosensitizing effect on them. Presumably, the above-described biological activity of CeF_3_ NPs can be realized by their effect on the modulation of various intracellular molecular cascades that differ in normal and tumor cells.

The effect of ionizing irradiation on eukaryotic cells leads to disruption of the electron transport chain system in mitochondria, which leads to delayed radiation-induced MMP collapse and increased generation of reactive oxygen species (ROS) [36,37]. Increased generation of ROS in mitochondria and a decrease in ATP synthesis in irradiated cells can be observed for a long time [38]. Thus, damaged mitochondria become inducers of oxidative stress over a long period of time. Using the cationic dye tetramethylrhodamine (TMRE), which accumulates in mitochondria in a voltage-dependent manner, we analyzed the mitochondrial membrane potential (MMP) of cell cultures 72 h after irradiation (Figure 3). It was revealed that irradiation of hMSC at a dose of 15 Gy does not lead to a significant decrease in MMP (Figure 3a). Pretreatment of cells with CeF_3_ NPs also does not affect the MMP level after irradiation. It is worth noting that after irradiation we observed morphological changes, in particular, the enlargement of mitochondria, which confirms the development of the senescence process. Pretreatment of cancer cells with CeF_3_ NPs leads to a slight decrease in the level of MMP, but no statistically significant decrease is observed. Irradiation of tumor cells leads to hyperpolarization of mitochondria (increase in MMP), and their pretreatment with CeF_3_ NPs nanoparticles only enhances this effect in a dose-dependent manner (Figure 3b). Thus, the combined effect of CeF_3_ NPs and X-ray irradiation have exactly the opposite effect on the mitochondrial membrane potential of tumor cells.

Ionizing radiation generally leads to diverse DNA damages, including single-strand and double-strand breaks [3]. Double-strand breaks (DSBs) make up a small part of such damage, while it is their number that is decisive for the further fate of the cell including the cell cycle arrest, DNA-repair processes activation or apoptosis triggering [39]. Using γ-H2AX histone staining with FITC-labeling antibodies, we evaluated the dynamics of the DSBs repair in normal and cancer cells after irradiation. Figure 4 shows that the number of double-strand breaks significantly reduced in the presence of CeF_3_ NPs in both normal (hMSC) and cancer (MCF-7) cells 1 h after irradiation. Then, after 4 h of incubation, the dynamics of the repair of DSBs become different for normal and cancer cells. For normal cells, the protective effect of CeF_3_ NPs is observed for all CeF_3_ NPs concentrations, which is expressed in a decrease in the number of DSBs (Figure 4a,c). Meanwhile, for MCF-7cancer cells, there is a significant increase in DSBs in comparison with the irradiated control (Figure 4b,d).

The effect of CeF_3_ NPs on the expression pattern in both cancer (MCF-7) and normal (hMSC) cells revealed downregulation of most genes (49 out of 89) (Figure 5). For example, genes coding some members of the glutathione peroxidase family (GPX1, GPX2, GPX5) were significantly down-regulated. Irradiation of MCF-7 cells in the presence of CeF_3_ NPs revealed upregulation in 13 genes, as well as suppressed expression of 17 studied genes. PCA analysis revealed a clear effect of radiation on the pattern of gene expression, and its pronounced modulation in the presence of CeF_3_ NPs. It should be noted that the separate action of irradiation or CeF_3_ NPs similarly influenced the expression patterns for both types of cell cultures, while the combined action of CeF_3_ NPs and X-ray irradiation was fundamentally different for cancer and normal cells. This difference can be associated with the effect of CeF_3_ NPs on the signaling pathways of the cell, in particular, on NF-κB. NF-κB is a transcription factor that participates in the regulation of various physiological processes, including response to oxidative stress, inflammation and apoptosis [40]. The activation mechanisms of the NF-κB pathway are associated with inhibition of the IκB subunit. Previously, it was shown that CeO_2_ NPs decrease intracellular oxidative stress and suppress phosphorylation of IκBα and the translocation of P65 subunits of NF-κB into the cell nuclei, which provide cell protection under cigarette smoke exposure [41].

Thus, we hypothesized that CeF_3_ NPs are able to modulate the activity of NF-κB both in the direction of activation and in the direction of suppression. The NF-κB signaling pathway is also responsible for the repair of DSBs after X-ray exposure [42]. In cancer cell (MCF-7 cells), inhibition of the IKK-NFκB pathway by a specific inhibitor of IKKβ was found to significantly reduce the repair rate of irradiation-induced DSBs and sensitized MCF-7 cells to clonogenic cell death. Recent studies show that CeO_2_ NPs are able to influence key cell signaling pathways, regulating the processes of proliferation, apoptosis, differentiation, protein synthesis and cell cycle progression [43]. Thus, the study of cerium-containing nanoparticles effect on cell signaling pathways could be one of the ways to increase the sensitization of cancer cells to ionizing radiation, increasing the efficiency of radiation therapy of oncological diseases.

Cerium-containing nanoparticles are considered as one of the most promising redox-active agents for such purposes, since they can act both as a prooxidant and as an antioxidant, depending on the microenvironment conditions [44]. We proposed a hypothetical scheme for the activity of CeF_3_ NPs in normal and cancer cells (Figure 6), which describes the possible molecular mechanisms of cell response under X-ray irradiation. Prooxidant ROS-mediated tumor therapy is considered to be one of the strategies to increase the effectiveness of treatment [45]. The mechanism of generation of excess ROS can be used to treat cancer by inducing cytotoxicity, by introducing a prooxidant agent to raise the intracellular ROS level above the cytotoxic threshold. It is known that the metabolism rate, proliferative activity and ROS levels in tumor cells are much higher than in normal ones [46]. Basal levels of ROS in cancer cells are initially elevated and the antioxidant defense system is disrupted, whereas the maintenance of homeostatic basal ROS levels in normal cells is modulated by the antioxidant defense system at a normal level.

Most authors explain the therapeutic effects of cerium-containing nanoparticles through the redox switching of Ce^3+^ ↔ Ce^4+^. At the same time, it is noted that pH plays a key role in the therapeutic effectiveness of cerium-containing nanoparticles. CeO_2_ nanoparticles exhibit superoxide dismutase-like activity under neutral and acidic conditions, which can catalyze the dismutation of superoxide radicals (O_2_^−^) into H_2_O_2_. Interestingly, CeO_2_ nanoparticles are highly active when decomposing H_2_O_2_ in a neutral medium but are inert under acidic conditions. Under acidic conditions, an excess of H^+^ can inhibit the conversion of Ce^4+^ to Ce^3+^, which catalyzes the decomposition of the absorbed surface of H_2_O_2_, which, in turn, disrupts the repeated action of active catalytic centers and blocks antioxidant cycles. Thus, high levels of H_2_O_2_ accumulate in cancer cells, which leads to their death. Considering that X-ray irradiation induces radiolysis of water and the formation of a large number of different types of ROS and RNS, which increase the load on the antioxidant system, CeO_2_ nanoparticles enhance the damaging effect under irradiation. Meanwhile, this mechanism or radiation-induced sensitization cannot be the only possible explanation for the selective therapeutic effect on tumor cells, since the relative importance of other factors, in particular Ce dissolution and the role of anions in microenvironment, remain poorly understood.

It is very interesting that CeO_2_ nanoparticles are able to selectively protect cells depending on the radiation energy used [47]. In particular, it was previously shown that CeO_2_ nanoparticles significantly reduce the effectiveness of radioprotection of cells exposed at an X-ray irradiation energy of 150 kVp compared with irradiation with a voltage of 10 MV. The authors attribute this change in efficiency to a significant increase in the generation of Auger electrons with high linear energy transfer under low-energy irradiation. Thus, it is necessary to take into account these features when using cerium-containing drugs as a radioprotective agent in the framework of radiation therapy. Given such an unexpected contribution of radiation intensity, it is possible to adjust the irradiation conditions to obtain the necessary therapeutic effect when using cerium-containing radioprotectors/radiosensitizers, including such a promising and actively developing approach in tumor radiotherapy as FLASH-therapy [48].

Thus, cerium-containing nanoparticles are a promising multimodal customizable platform for various radiotherapy purposes.

## 4. Conclusions

CeF_3_ NPs significantly reduce the content of hydrogen peroxide and hydroxyl radicals after X-ray exposure. CeF_3_ NPs in hMSC act as a proliferation stimulator and a radioprotector, reducing the number of DSBs, maintaining high viability, proliferative activity and mitochondrial membrane potential, and maintaining viability, proliferative activity and mitochondrial membrane potential close to control values after irradiation. Meanwhile, in MCF-7 cells CeF_3_ NPs act as proliferation suppressor and a radiosensitizer, which slows down the rate of DSBs repair 4 h after irradiation. CeF_3_ NPs reduce the proliferative activity of MCF-7 cells, leading to radiation-induced hyperpolarization of mitochondria. The molecular mechanisms of the selective action of CeF_3_ NPs are most likely associated with the modulation of intracellular signaling pathways through the regulation of the redox status of the cell, which we confirmed by qPCR analysis. The observed selective enhancement of DSBs repair process of cerium-containing nanoparticles can be used to increase the sensitization of cancer cells in a patient during radiation therapy and to protect healthy tissues in the immediate vicinity of the tumor.

## Figures and Tables

**Figure 1 biomedicines-12-00011-f001:**
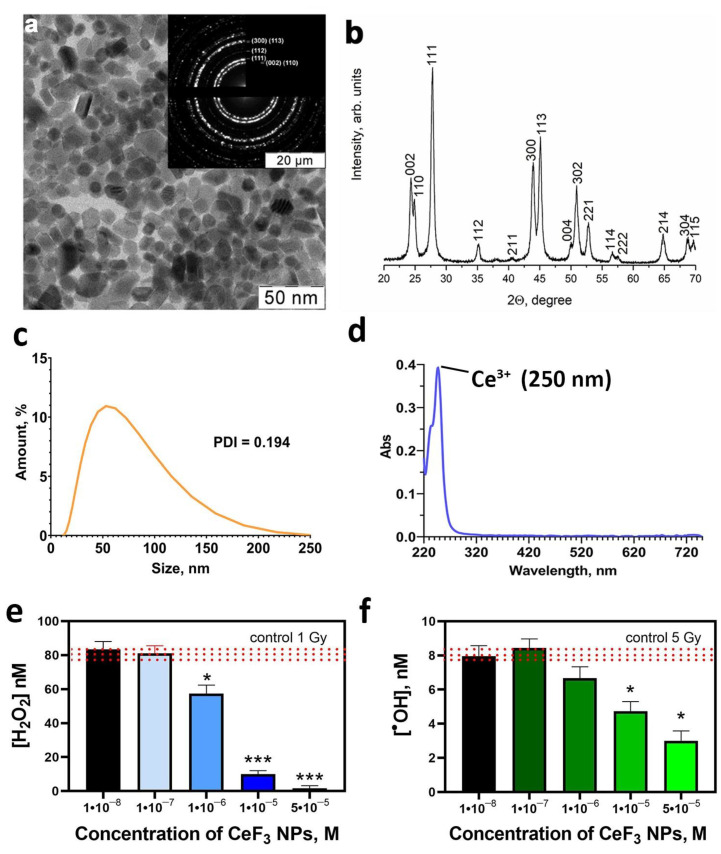
The TEM image (**a**), XRD (**b**), hydrodynamic radii distribution (**c**) and UV spectra (**d**) of CeF_3_ NPs; hydrogen peroxide (**e**) and hydroxyl radicals (**f**) formation induced by X-ray irradiation in a solution, containing CeF_3_ NPs. Insets in (**a**) show selected area electron diffraction data. Significant differences compared with untreated control using the *t*-test, * *p* < 0.05, *** *p* < 0.005. The red line shows the value for the irradiated solution without CeF_3_ NPs (irradiated control).

**Figure 2 biomedicines-12-00011-f002:**
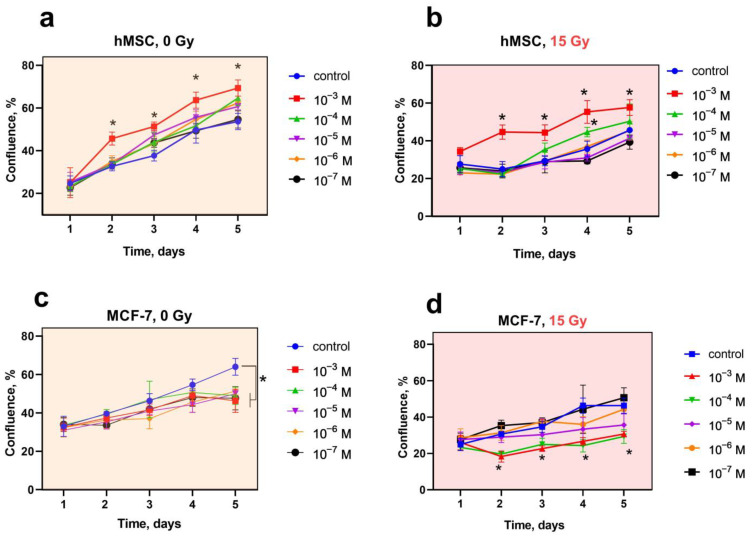
(**a**–**d**) The cell proliferation analysis of hMSC and MCF-7 cell lines after X-ray irradiation with CeF_3_ NPs. The hMSC and MCF-7 cells were overnight pretreated with CeF_3_ NPs (10^−3^–10^−7^ M) and then irradiated with 15 Gy of X-ray. The change in the confluence of cell cultures was observed for 5 days. Significant differences compared with untreated control using the *t*-test, * *p* < 0.05.

**Figure 3 biomedicines-12-00011-f003:**
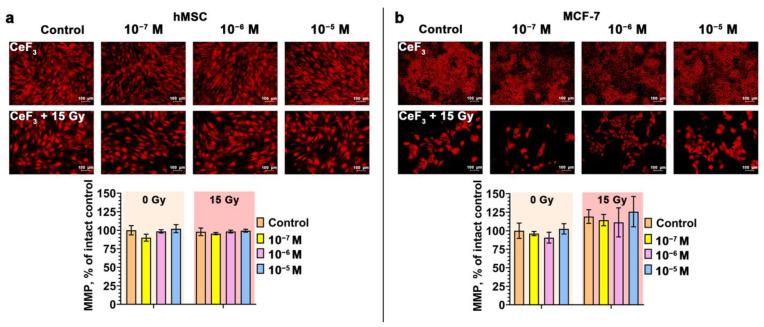
(**a,b**) Mitochondrial membrane potential (MMP) of hMSC and MCF-7 cells 72 h after X-ray irradiation (15 Gy dose) with CeF_3_ NPs in various concentrations (10^−7^–10^−5^ M). The MMP of cells are indicated as the mean with standard deviation (SD) (*n* = 3).

**Figure 4 biomedicines-12-00011-f004:**
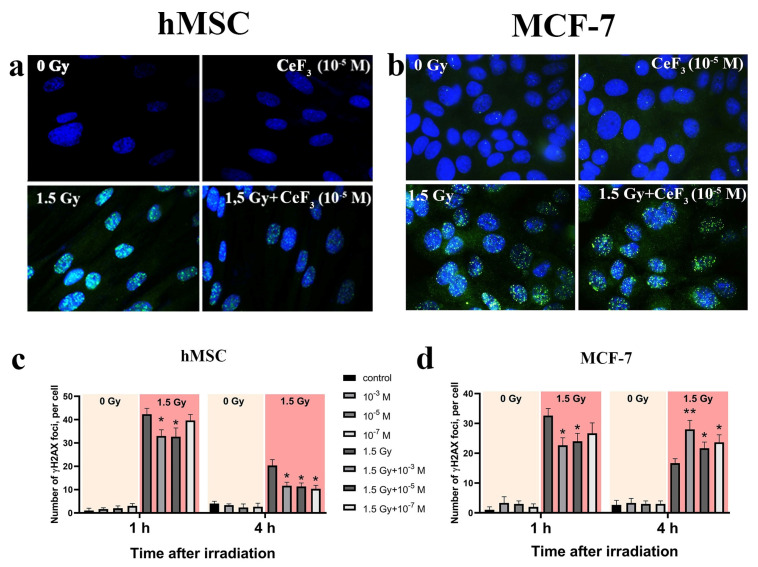
Radiation-induced γ-H2AX foci determined using FITC-labeled antibody by fluorescent microscopy (**a**,**b**). Quantitative analysis of DSBs repair process 1 h or 4 h after irradiation (**c**,**d**). The hMSC and MCF-7 cells were pretreated with CeF_3_ NPs irradiated with 1.5 Gy of X-ray and let to recover for 1 h or 4 h post irradiation. Data points represent means of two experiments, at least 200 cells were counted at each treatment condition (**b**). Unirradiated cells (0 Gy) were used as control. Data presented as mean of γ-H2AX foci per cell ± SD, *n* = 200. Statistically significant differences are indicated by the * *p* < 0.05, ** *p* < 0.01.

**Figure 5 biomedicines-12-00011-f005:**
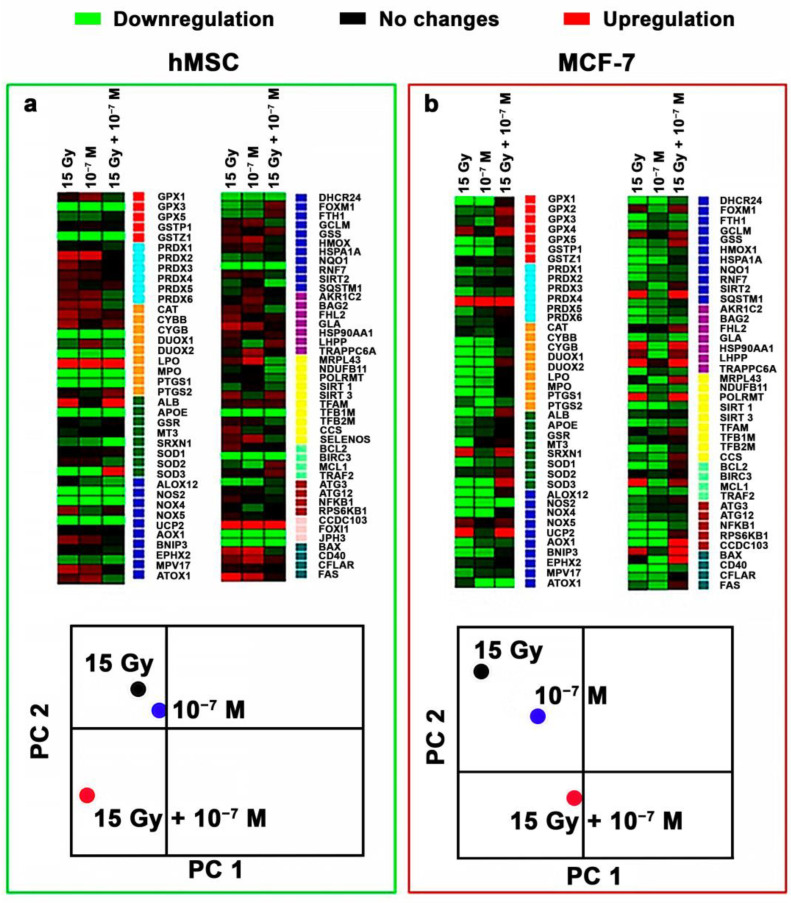
Heat map of gene expression in hMSC (**a**) and MCF-7 (**b**) cells treated with CeF_3_ NPs (10^−7^ M). The intensity scale of the standardized expression values ranges from −3 (green: low expression) to +3 (red: high expression), with the 1:1 intensity value (black) representing the control (non-treated cells). Principal component analysis (PCA) data is presented in the heat map.

**Figure 6 biomedicines-12-00011-f006:**
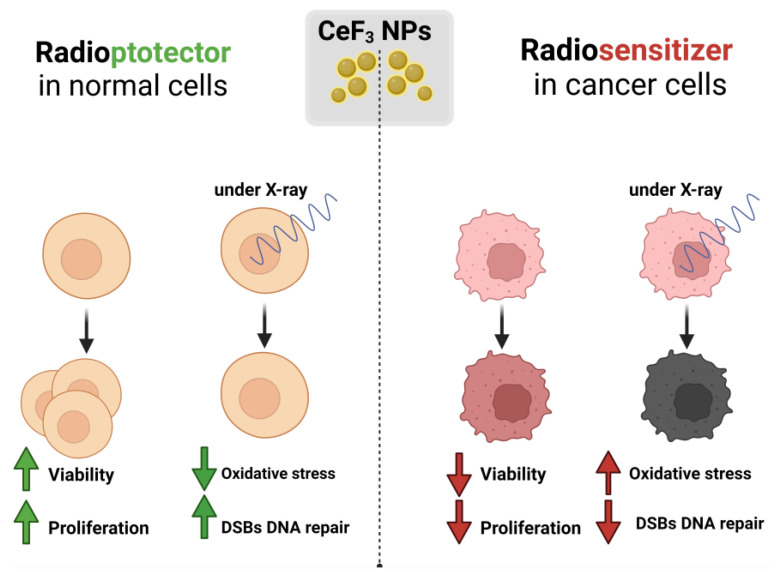
Schematic representation of the possible biological activity of CeF_3_ nanoparticles on normal and cancer cells under X-ray irradiation.

## Data Availability

The data presented in this study are available in article.

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
