# Peer review of "Redox-Active Cerium Fluoride Nanoparticles Selectively Modulate Cellular Response against X-ray Irradiation In Vitro"

_biomedicines, 2023, doi:10.3390/biomedicines12010011_

Round 1
Reviewer 1 Report
Comments and Suggestions for Authors
In this study, the authors developed the CeF3 NPs to achieve the selective protecting of normal cells from radiation-induced injuries. The CeF3 NPs provided a significant decrease in the number of DSBs in normal cells, while in tumor cells the radiation damage dramatically increased in the presence of CeF3 NPs after X-rays irradiation. The manuscript is clear and reasonable logic structure, and the results support the conclusions. In order to be published in Biomedicines, the following minor issues should be addressed.
1. The number of DSBs significantly reduced in the presence of CeF3 NPs in hMSCs cell at 4 hours after irradiation. Meanwhile, for cancer cells, there is a significant increase of DSBs in comparison with the irradiated control. Only one type of normal cell and cancer cell was detected in the manuscript, other normal cells or cancer cell lines need to be added to verify the universality of the conclusion.
2. The authors should detect the valence change of cerium ions after the reaction of CeF3 and H2O2, which is conducive to revealing the mechanism of scavenging ROS by CeF3 NPs.
3. There are two Figures 4 in the manuscript (Pages 8 and 10), the Figure 4 on page 10 should be Figure 5, please check them carefully.
4. Please unify the format of X-ray in the manuscript, X-ray or X-rays?
5. Please unify the format of human mesenchymal stem cells (hMSc) in the manuscript, hMSc or hMSCs?
6. Please check the format of all References much more carefully, the valence state of ions needs to be superscript, and the number of ions needs to be subscript (Reference 16, 19, 21, 22, 23, 24, 30, 31, 34).
Comments on the Quality of English LanguageMinor editing of English language required
Author Response
We are extremely grateful for the reviewers’ comments that are aimed at improving our paper. We have thoroughly revised the manuscript in accordance with the reviewers’ comments. We have carefully checked all the points and tried to take into account all the questions and suggestions.
Reviewer #1
General comment: In this study, the authors developed the CeF3 NPs to achieve the selective protecting of normal cells from radiation-induced injuries. The CeF3 NPs provided a significant decrease in the number of DSBs in normal cells, while in tumor cells the radiation damage dramatically increased in the presence of CeF3 NPs after X-rays irradiation. The manuscript is clear and reasonable logic structure, and the results support the conclusions. In order to be published in Biomedicines, the following minor issues should be addressed.
Discussion: We thank the reviewer for the positive evaluation of our work.
Issue 1: The number of DSBs significantly reduced in the presence of CeF3 NPs in hMSCs cell at 4 hours after irradiation. Meanwhile, for cancer cells, there is a significant increase of DSBs in comparison with the irradiated control. Only one type of normal cell and cancer cell was detected in the manuscript, other normal cells or cancer cell lines need to be added to verify the universality of the conclusion.
Discussion: We thank the reviewer for this comment. Indeed, application of the terms «normal cells» and «tumor cells» generalizes conclusions drawn in this work. In the Abstract and Conclusion, we have changed the terms «normal cell» and «tumor cells» to hMSc and MCF-7 cell lines, respectively.
Changes made in the manuscript: Updated Abstract and Conclusion: the terms «normal cells» and «tumor cells» have been changed to hMSc and MCF-7 cell lines, respectively.
Issue 2: The authors should detect the valence change of cerium ions after the reaction of CeF3 and H2O2, which is conducive to revealing the mechanism of scavenging ROS by CeF3 NPs.
Discussion: We thank the reviewer for the valuable comment. We have performed an additional experiment to reveal changes in the cerium ions valence after the reaction of CeF3 NPs and H2O2. We have added the appropriate diagram of the CeF3 NPs UV- spectrum before and after the reaction with H2O2, indicating the cerium valence state to the Supplementary materials (See Figure S1)
Changes made in the abstract: Updated Supplementary materials: «Figure S1. Absorbance spectrum of CeF3 NPs before and after the reaction with H2O2» has been added to the manuscript Supplementary materials.
Figure S1. UV spectrum of CeF3 NPs before and after the reaction with H2O2. The absorbance spectrum shows absorption maxima of Ce3+ and Ce4+ ions in CeF3 NPs.
Issue 3: There are two Figures 4 in the manuscript (Pages 8 and 10), the Figure 4 on page 10 should be Figure 5, please check them carefully.
Discussion: We thank the reviewer for this comment. We have changed caption to the illustration «Heat map of gene expression in hMSc (a) and MCF-7 (b) cells treated with CeF3 NPs (10-7 M)» from Figure 4 to Figure 5.
Changes made in the manuscript: Updated Results and Discussion: Caption to the illustration «Heat map of gene expression in hMSc (a) and MCF-7 (b) cells treated with CeF3 NPs (10-7 M)» has been changed from Figure 4 to Figure 5.
Issue 4: Please unify the format of X-ray in the manuscript, X-ray or X-rays?
Discussion: We thank the reviewer for this comment. We have unified the format of X-ray with the «X-ray» format in the manuscript.
Changes made in the manuscript: Updated manuscript: All «X-rays», «X rays» and «X ray» have been replaced by «X-ray».
Issue 5: Please unify the format of human mesenchymal stem cells (hMSc) in the manuscript, hMSc or hMSCs?
Discussion: We thank the reviewer for this comment. We have unified the format of human mesenchymal stem cells (hMSС) with the «hMSС» format in the manuscript.
Changes made in the manuscript: Updated manuscript: All «hMSc» have been replaced by «hMSС».
Issue 6: Please check the format of all References much more carefully, the valence state of ions needs to be superscript, and the number of ions needs to be subscript (Reference 16, 19, 21, 22, 23, 24, 30, 31, 34).
Discussion: We thank the reviewer for this comment. We have checked and fixed superscript and subscript in the Reference.
Changes made in the manuscript: Updated Reference. All of the valence state of ions have been formatted to superscript as well as all of the number of ions have been formatted to subscript.
Reviewer 2 Report
Comments and Suggestions for Authors
This study utilized redox-active cerium fluoride nanoparticles in conjunction with radiotherapy to target tumors. The findings indicate that the introduction of CeF3NPs can safeguard normal cells while effectively managing and eliminating cancerous cells. However, several concerns arise regarding this research:
1. Title: The title should be reconsidered to reflect the focus on cell study within radiotherapy.
2. Introduction: Incorporating recent advancements in nanomaterials within cancer therapy, referencing works such as Chow (doi.org/10.1016/B978-0-12-823152-4.00001-6) would enhance the introduction.
3. Introduction: An explanation is needed regarding why the protective mechanism occurs in normal cells but not in cancer cells, as well as why the cell-killing mechanism occurs specifically in cancer cells under irradiation.
4. A schematic diagram illustrating the differential interaction of CeF3NPs with normal and cancer cells would help clarify the distinct mechanisms involved.
5. Section 2.3: Considering that human radiotherapy generally uses MV photon beams rather than kV beams, discussing how the beam energy might impact particle interaction and the cell-killing/protection mechanisms in the study is crucial.
6. Discussing the dose rate effect on the antioxidant activity of the nanoparticles is recommended. Notably, acknowledging FLASH radiotherapy as an innovative technique with an ultra-high dose rate of ionizing beams (Siddique et al, Cancers 2023;15:3883) would be beneficial.
7. A significant drawback of this work is the absence of preclinical experimental evidence supporting the application of CeF3NPs. The focus solely on individual normal and cancer cells in cell experiments neglects the necessity of a tumor structure containing both cell types interconnected for a more comprehensive analysis.
Comments on the Quality of English LanguageI have no problem with the English of the paper.
Author Response
Reviewer #2
We are extremely grateful for the reviewers’ comments that are aimed at improving our paper. We have thoroughly revised the manuscript in accordance with the reviewers’ comments. We have carefully checked all the points and tried to take into account all the questions and suggestions.
General comment: This study utilized redox-active cerium fluoride nanoparticles in conjunction with radiotherapy to target tumors. The findings indicate that the introduction of CeF3 NPs can safeguard normal cells while effectively managing and eliminating cancerous cells. However, several concerns arise regarding this research.
Discussion: We thank the reviewer for the positive evaluation of our work.
Issue 1: The title should be reconsidered to reflect the focus on cell study within radiotherapy.
Discussion: We thank the reviewer for this comment. We have reconsidered the title of the article and changed it.
Changes made in the manuscript: New title: «Redox-active cerium fluoride nanoparticles selectively modulate cellular response against X-ray irradiation in vitro».
Issue 2: Introduction: Incorporating recent advancements in nanomaterials within cancer therapy, referencing works such as Chow (doi.org/10.1016/B978-0-12-823152-4.00001-6) would enhance the introduction.
Discussion: We thank the reviewer for this comment.
Changes made in the manuscript: We have added this this relevant work to the introduction section.
Issue 3: Introduction: An explanation is needed regarding why the protective mechanism occurs in normal cells but not in cancer cells, as well as why the cell-killing mechanism occurs specifically in cancer cells under irradiation.
Discussion: We thank the reviewer for this comment. We have added a discussion of the mechanism of selective action of cerium-containing nanoparticles.
Changes made in the manuscript: Prooxidant ROS–mediated tumor therapy is considered as one of the strategies to increase the effectiveness of treatment [44]. The mechanism of generation of excess ROS can be used to treat cancer by inducing cytotoxicity, by introducing a prooxidant agent to raise the intracellular ROS level above the cytotoxic threshold. It is known that the level of metabolism, proliferative activity and ROS levels in tumor cells are much higher than in normal ones [45]. Basal levels of ROS in cancer cells are initially elevated and the antioxidant defense system is disrupted, whereas the maintenance of homeostatic basal ROS levels in normal cells is modulated at a normal level by the antioxidant defense system. Cerium-containing nanoparticles are considered as one of the most promising redox-active agents for such purposes, since they can act both as a prooxidant and as an antioxidant, depending on the microenvironment conditions [46]. Most authors explain the therapeutic effects of cerium-containing nanoparticles through the redox switching of Ce3+ ↔ Ce4+. At the same time, it is believed that pH plays a key role in the therapeutic effectiveness of cerium-containing nanoparticles. CeO2 nanoparticles exhibit superoxide dismutase-like activity under neutral and acidic conditions, which can catalyze the dismutation of superoxide radicals (·O2-) into H2O2. Interestingly, CeO2 nanoparticles are highly active when decomposing H2O2 in a neutral medium, but are inert under acidic conditions. Under acidic conditions, an excess of H+ can inhibit the conversion of Ce4+ to Ce3+, which catalyzes the decomposition of the absorbed surface of H2O2, which, in turn, disrupts the repeated action of active catalytic centers and blocks antioxidant cycles. Thus, high levels of H2O2 accumulate in cancer cells, which leads to their death. Considering that radiation induces radiolysis of water and the formation of a large number of different types of ROS and RNS, which increase the load on the antioxidant system, and CeO2 nanoparticles enhance the damaging effect under irradiation conditions. Meanwhile, this mechanism cannot be the only possible explanation for the selective therapeutic effect on tumor cells, since the relative importance of other factors, in particular Ce dissolution and the role of anions in microenvironment remain poorly understood.
Issue 4: A schematic diagram illustrating the differential interaction of CeF3 NPs with normal and cancer cells would help clarify the distinct mechanisms involved.
Discussion: We thank the reviewer for this comment. We have added a schematic representation of the effect of CeF3 nanoparticles on normal and cancer cells.
Changes made in the manuscript: Figure 6
Issue 5: Section 2.3: Considering that human radiotherapy generally uses MV photon beams rather than kV beams, discussing how the beam energy might impact particle interaction and the cell-killing/protection mechanisms in the study is crucial.
Discussion: We thank the reviewer for this comment. We have added to the discussion section data on the effect of radiation energy on the effectiveness of radioprotection/radiosensitization of cerium-containing nanoparticles.
Changes made in the manuscript:
It is very interesting that CeO2 nanoparticles are able to selectively protect cells depending on the radiation energy used [47]. In particular, it was previously shown that CeO2 nanoparticles significantly reduce the effectiveness of radioprotection of cells exposed at an X-ray irradiation energy of 150 kVp compared with irradiation with a voltage of 10 MV. The authors attribute this change in efficiency to a significant increase in the generation of Auger electrons with high linear energy transfer under low-energy irradiation. Thus, it is necessary to take into account these features when using cerium-containing drugs as a radioprotective agent in the framework of radiation therapy.
Issue 6: Discussing the dose rate effect on the antioxidant activity of the nanoparticles is recommended. Notably, acknowledging FLASH radiotherapy as an innovative technique with an ultra-high dose rate of ionizing beams (Siddique et al, Cancers 2023;15:3883) would be beneficial.
Discussion: We thank the reviewer for this comment. We have added information about many promising approaches to radiotherapy for resistant cells such as flash therapy.
Changes made in the manuscript: Given such an unexpected contribution of radiation intensity, it is possible to adjust the irradiation conditions to obtain the necessary therapeutic effect when using cerium-containing radioprotectors/radiosensitizers, including such a promising and actively developing approach in tumor radiotherapy as flash therapy [48].
Issue 7: A significant drawback of this work is the absence of preclinical experimental evidence supporting the application of CeF3 NPs. The focus solely on individual normal and cancer cells in cell experiments neglects the necessity of a tumor structure containing both cell types interconnected for a more comprehensive analysis.
Discussion: We thank the reviewer for this comment. We agree with the reviewer that experimental confirmation in vivo will allow us to more deeply reveal the potential of this type of cerium-containing nanoparticles as a promising radioprotector, however, we have not set such goals in this work. This article is the first to demonstrate the fundamental possibility of using cerium fluoride nanoparticles as a selective radiosensitizer/radiosensitizer. In the future, we plan to conduct experiments in vivo tumor models to study the radiosensitizing effect of cerium fluoride nanoparticles upon their intratumoral administration.
Changes made in the manuscript: No changes were made
Round 2
Reviewer 2 Report
Comments and Suggestions for Authors
I am satisfied with the modifications from the authors as per my comments. The presentation and quality of this manuscript are improved a lot. I have no further questions.
Comments on the Quality of English LanguageI have no problem to read and understand this revised manuscript.